# Insight into the Role of Conductive Polypyrrole Coated on Rice Husk-Derived Nanosilica-Reduced Graphene Oxide as the Anodes: Electrochemical Improvement in Sustainable Lithium-Ion Batteries

**DOI:** 10.3390/polym15244638

**Published:** 2023-12-07

**Authors:** Natthakan Ratsameetammajak, Thanapat Autthawong, Kittiched Khunpakdee, Mitsutaka Haruta, Torranin Chairuangsri, Thapanee Sarakonsri

**Affiliations:** 1Department of Chemistry, Faculty of Science, Chiang Mai University, Chiang Mai 50200, Thailand; natthakan.ntr@gmail.com (N.R.); thanapat.chem@gmail.com (T.A.); kittiched_kh@cmu.ac.th (K.K.); 2Center of Excellent for Innovation in Chemistry (PERCH-CIC), Faculty of Science, Chiang Mai University, Chiang Mai 50200, Thailand; 3Materials Science Research Center, Faculty of Science, Chiang Mai University, Chiang Mai 50200, Thailand; 4Office of Research Administration, Chiang Mai University, Chiang Mai 50200, Thailand; 5Institute for Chemical Research, Kyoto University, Kyoto 611-0011, Japan; haruta@eels.kuicr.kyoto-u.ac.jp; 6Department of Industrial Chemistry, Faculty of Science, Chiang Mai University, Chiang Mai 50200, Thailand; torranin.c@cmu.ac.th

**Keywords:** conductive polypyrrole, nanocomposites, anode materials, lithium-ion batteries

## Abstract

Polypyrrole (PPy) is a type of conducting polymer that has garnered attention as a potential electrode material for sustainable energy storage devices. This is mostly attributed to its mechanical flexibility, ease of processing, and ecologically friendly nature. Here, a polypyrrole-coated rice husk-derived nanosilica-reduced graphene oxide nanocomposite (SiO_2_-rGO@PPy) as an anode material was developed by a simple composite technique followed by an in situ polymerization process. The architecture of reduced graphene oxide offers a larger electrode/electrolyte interface to promote charge-transfer reactions and provides sufficient space to buffer a large volume expansion of SiO_2_, maintaining the mechanical integrity of the overall electrode during the lithiation/delithiation process. Moreover, the conducting polymer coating not only improves the capacity of SiO_2_, but also suppresses the volume expansion and rapid capacity fading caused by serious pulverization. The present anode material shows a remarkable specific reversible capacity of 523 mAh g^−1^ at 100 mA g^−1^ current density and exhibits exceptional discharge rate capability. The cycling stability at a current density of 100 mA g^−1^ shows 81.6% capacity retention and high Coulombic efficiency after 250 charge–discharge cycles. The study also pointed out that this method might be able to be used on a large scale in the lithium-ion battery industry, which could have a big effect on its long-term viability. Creating sustainable nanocomposites is an exciting area of research that could help solve some of the biggest problems with lithium-ion batteries, like how easy they are to make and how big they can be used in industry. This is because they are sustainable and have less of an impact on the environment.

## 1. Introduction

As the commercialization of lithium-ion batteries continues to advance, rechargeable lithium-ion batteries are currently leading prospects for a wide range of technical applications, including portable electronic gadgets and electric vehicles. As a key component, the electrochemical properties of lithium-ion batteries are largely determined by the electrode material. SiO_2_-based materials are among the most promising anode materials for Li-ion batteries (LIBs) due to their low discharge voltage (0.2 V vs. Li^+^/Li), low cost, and global abundance [1,2]. It is vital to establish SiO_2_ from biomass rich in lignocellulose, such as rice husk [1,3,4], bamboo leaf [5], and sugarcane [6]. Consequently, it is imperative to establish biomass-derived SiO_2_, which is accomplished by transforming waste into products with added value and is regarded as an environmentally friendly preparation method. SiO_2_ has a theoretical specific capacity of 1965 mAh g^−1^, which is considerably greater than the graphite anode materials that are currently in use (372 mAh g^−1^) [7,8,9]. However, the hundred percent volume expansion of SiO_2_ particles during charge–discharge cycles results in rapid capacity degradation, severely limiting its practical application [10].

To overcome this issue, substantial research has been conducted on nanostructured SiO_2_ materials with the purpose of improving the electrochemical performance of anodes for LIBs. In addition, polymers with intrinsic conductivity have lately attracted considerable interest as anode materials for enhancing LIB performance [11]. Conducting polymers (CPs) have a distinctive structural feature, wherein their polymer backbone is composed of a sequence of alternating single and double bonds. The system of delocalized π-electrons is generated by the overlapping *p*-orbitals, resulting in optical and electrical features that are both fascinating and valuable [12]. According to recent research, it has been shown that these materials generally demonstrate an electrical conductivity that varies within the range of 0.01 to 500 S cm^−1^ [13]. Additionally, there has been a growing interest in electrochemical energy storage devices that possess qualities like strength, flexibility, low production costs, low self-discharge rates, tolerance to overdischarge, and a long cycle life [14]. Polyheterocycles, including polypyrrole (PPy), polyaniline (PANI), polythiophene (PTh), poly(3,4-ethylenedioxythiophene) (PEDOT), and their derivatives [15], are considered to be highly intriguing conducting polymers (CPs) in the context of battery applications. These materials have been the subject of investigation due to their exceptional stability in atmospheric conditions and favorable electrochemical characteristics. PPy has garnered significant interest within the field of conductive polymers because of its exceptional attributes. These include its comparatively affordable cost, remarkable redox capabilities, commendable electrical conductivity, biocompatibility, and chemical stability [16,17,18].

In terms of enhancing conductivity, the polymer matrix can alleviate the internal stress of solid particle anodes that experience a significant volume change during charge–discharge [19]. A number of anode materials, such as Si-PPy [20,21], SiO_2_-PPy [22], SnO_2_-PPy [23], have recently shown massively better cycling performance in comparison to standalone solid particles. Polypyrrole has been demonstrated in these composites as a surface coating layer and/or active matrix for usage in batteries to enhance structural stability and cycle performance [9,22,24,25]. Due to the polymer’s ability to form a conducting matrix, which provides a conducting backbone for the particles, it significantly enhances the conductivity of the electrode by decreasing the particle-to-particle contact resistance and preventing unwanted side reactions between active materials and electrolytes [26].

In contrast, SiO_2_ composites having a buffer layer, such as carbon materials, displayed better stability due to their high overall electrical conductivity and their capacity to reduce the internal stress of SiO_2_ [27,28,29]. Graphene, particularly when chemically functionalized, can enhance the storage and electric charge of batteries and supercapacitors [30,31,32,33]. For these reasons, it has considerable potential as a material for advanced energy storage technologies [34,35]. Specifically, reduced graphene oxide (rGO), the distribution of carbon atoms at a 2D monolayer resembling a honeycomb, is typically substituted for graphene, primarily because it can be generated in vast quantities via the massive chemical reduction of graphene oxide. Because it can ensure rapid electron transport to the electrode support and Li^+^ ion storage, it has shown great electrochemical performance. During short-term charge/discharge cycling (~50 cycles), single graphene anodes lose more than 50% of their capacity irreversibly. Solid nanoparticles must be introduced between these graphene layers to overcome the re-stacking issue.

Even though substantial progress has been made in mitigating the volume expansion of SiO_2_ by integrating conducting polymers or graphene into binary systems, there is still ample scope for improvement in the electrochemical performance of SiO_2_ anodes. We present a feasible ternary nanocomposite of SiO_2_-rGO@PPy anode material comprised of nano-SiO_2_ nanoparticles derived from rice husk that are surrounded by sheets of reduced graphene oxide (rGO) and a thin layer of conductive polypyrrole (PPy). In addition to the likely synergistic effect of rice husk-derived nano-SiO_2_ nanoparticle structure and PPy buffer layers, the two-dimensional graphene not only increases the strength of the electrode, but it also significantly improves electron and Li^+^ mobility in the electrode. Comparing the ternary nanocomposite anode to the single PPy and binary SiO_2_-rGO anode systems, the ternary nanocomposite anode exhibited a superior cycle stability and rate performance.

## 2. Materials and Methods

### 2.1. Materials

The rice husks were acquired from a local agriculturist in northern Thailand. We purchased graphite (powder, <20 µm, synthetic) and the pyrrole monomer (C_4_H_4_NH, 98%) from Sigma-Aldrich Co., Ltd (St. Louis, MO, USA). Iron (III) chloride hexahydrate (FeCl_3_·6H_2_O, 98%) and potassium permanganate (KMnO_4_, 99%) were purchased from KEMAUS (Sydney, Australia). RCI-Labscan (Bangkok, Thailand) manufactured sodium hydroxide (NaOH, 99%) and hydrochloric acid (HCl, 37%). JTBaker (Phillipsburg, NJ, USA) was the vendor of sulfuric acid (H_2_SO_4_, 98%).

### 2.2. Synthesis of Rice Husk-Derived Nanosilica-Reduced Graphene Oxide@polypyrrole (SiO_2_-rGO@PPy)

The comprehensive synthesis route that leads to SiO_2_-rGO@PPy nanocomposites is illustrated in Figure 1. Nanosilica-reduced graphene oxide nanocomposites (SiO_2_@rGO) are produced from rice husk (the ratio of SiO_2_ to rGO is 30:70). The rice husks (RHs) obtained from agricultural waste were subjected to a rinsing process using water at room temperature in order to eliminate any mechanical impurities and dust. Subsequently, the RHs were dried at a temperature of 100 °C until they reached a state of total dryness. The RHs that had been rinsed and dried were immersed in an aqueous solution of HCl overnight, with the purpose of removing any metal impurities and promoting the hydrolysis of cellulose and hemicellulose. The sample was subsequently rinsed with water until a neutral pH of 7 was achieved, followed by drying at a temperature of 100 °C. Subsequently, the RHs underwent a two-stage burning process in a muffle furnace: first at a temperature of 500 °C for a duration of 2 h, followed by a subsequent burning at a temperature of 700 °C for a duration of 2 h. Initially, as-prepared silica was reprecipitated for 2 h at 120 °C in a NaOH solution. Graphene oxide (GO) was synthesized from graphite powder employing a modified Hummers’ method, followed by successive reduction within a tube furnace. Reduced graphene oxide (rGO) was synthesized by subjecting it to a five-hour heat treatment at a temperature of 800 °C under an inert atmosphere. The reduced graphene oxide was subsequently dispersed throughout the solution. After correcting the mixture’s pH to 7 with a hydrochloric acid solution, it was agitated overnight at room temperature. The substance was then centrifuged and washed with deionized water and ethanol. Afterwards, the samples were dried in a 60 °C hot air oven. PPy-coated SiO_2_@rGO nanocomposites were prepared using in situ polymerization. Initially, 75 mL of ethanol was mixed with 15 min of stirring to dissolve SiO_2_@rGO. Then, 0.01 mol of pyrrole monomer was slowly added to the solution while it was being agitated for 15 min. A suitable quantity of FeCl_3_·6H_2_O (10 mL in DI water) was applied drop-by-drop with magnetic stirring for 12 h to form a black precipitate. This black precipitate was meticulously and repeatedly washed with DI water and ethanol. Finally, the product was dried at 60 °C to obtain PPy-coated SiO_2_@rGO.

### 2.3. Morphological and Structural Characterization

The morphology of synthesized samples was examined using a scanning electron microscope (SEM, JEOL JSM-IT800, Tokyo, Japan). High-resolution transmission electron microscopy (HRTEM, JEOL JEM-2200FS, Tokyo, Japan) was used to analyze the microstructures of synthesized samples. Using energy-dispersive x-ray spectroscopy (EDX), elemental analysis was conducted with scanning transmission electron microscopy (STEM, JEOL JEM-2200F, Tokyo, Japan). Attenuated total reflectance: Fourier transform infrared (ATR-FTIR) spectra were acquired at room temperature with an ATR-FTIR spectrometer (Bruker, Tensor 27, Billerica, MA, USA). To evaluate the elemental composition, X-ray diffraction (XRD) (Rigaku Miniflex II desktop, Tokyo, Japan) measurements were performed. To determine the actual amount of each component in the nanocomposite, simultaneous thermal analysis (STA) was performed using a Rigaku (Thermo Plus Evo2, Tokyo, Japan) analyzer in air.

### 2.4. Electrochemical Measurements

The stainless-steel coin cell (CR2016) was assembled in an argon-filled glovebox using nanomaterials as anodes and lithium foil as a counter electrode, equipped with pure Li metal foil serving as the counter and reference electrodes. Copper foil was coated with a slurry mixture of prepared active materials, super P, and sodium alginate at a weight ratio of 70:15:15 to create the working electrode. The electrolyte was a LiPF_6_ solution in ethylene carbonate (EC). On an electrochemical workstation (PGSTAT 302N, Herisau, Switzerland), electrochemical impedance spectroscopy (EIS) was performed by applying an alternating current (AC) voltage of 2 mV in the frequency range from 100 kHz to 0.1 Hz. And cycle voltammetry measurements of the cells were taken at room temperature at a scan rate of 0.1 mV s^−1^ over the range of 0.1–3.0 V. Discharge–charge cycling was carried out at room temperature utilizing a battery test system (Neware BTS-4000, Shenzhen, China).

## 3. Results and Discussion

### 3.1. Characterization of SiO_2_-rGO@PPy Nanocomposite

The XRD patterns of as-prepared bare PPy, SiO_2_-rGO, and SiO_2_-rGO@PPy nanocomposites are compared in Figure 1a. XRD patterns of SiO_2_-rGO indicate two broad diffraction peaks at 2θ = 25.5° and 43.3°, which coincide with the diffraction patterns of the (002) and (100) crystallographic planes of the graphene materials (JCPDS no. 41-1487) [1,36]. In addition, the SiO_2_ in the SiO_2_-rGO nanocomposite displayed a broad peak at 2θ = 25.4, corresponding to amorphous silica (JCPDS no. 29-0085) [1,37,38]. The intensity increasing diffraction peaks of rGO and SiO_2_ at 2θ = 25° suggest that the SiO_2_ load was correctly synthesized. In accordance with the published literature, the XRD spectrum of bare PPy exhibited a 2θ value of 23°, indicating the amorphous nature of PPy [39,40]. Moreover, the XRD spectrum of the SiO_2_-rGO@PPy nanocomposites is almost similar to that of pure PPy and SiO_2_-rGO. Also, the peaks from SiO_2_-rGO@PPy nanocomposites have considerable overlapping peaks that cannot be distinguished by their phase; therefore, FTIR analysis was utilized to confirm the phase and chemical structure of the nanocomposites.

As indicated in Figure 1b, FTIR was also employed to analyze the chemical structure and determine the functional groups and bonding nature of bare PPy, SiO_2_-rGO, and SiO_2_-rGO@PPy. The samples of SiO_2_-rGO nanocomposites contain the following functional groups: the broad transmission at 1066 cm^−1^ is attributable to Si-O-C or Si-O-Si stretching vibration [1,41,42], while the peaks at 1541 cm^−1^ and 1040 cm^−1^ in the spectra of SiO_2_-rGO are ascribed to the stretching vibrations of the C=C and C-OH bands of rGO, respectively [43,44,45]. According to SiO_2_-rGO, the frequency of the peaks in the spectra of SiO_2_-rGO@PPy decreased, confirming the formation of the SiO_2_-rGO@PPy nanocomposite [46,47]. In addition, the distinctive peaks of SiO_2_-rGO@PPy are compatible with the PPy spectra below 1600 cm^−1^ [48]. PPy’s spectra reveal characteristic bands at 1541 cm^−1^ and 1456 cm^−1^, which correspond to the symmetrical stretching vibration of C=C on the pyrrole ring and the stretching mode of C-N vibrations, respectively [49,50]. Contrarily, a band at 1292 cm^−1^ is due to C-C stretching vibrations. The C-N stretching vibration of polypyrrole was another relatively intense band at 1157 cm^−1^. There was an intense peak at 1031 cm^−1^, which corresponds to the N-H in-plane deformation absorption of polypyrrole [51]. Low-intensity peaks at 773 cm^−1^ correspond to the distinctive peak of the pyrrole ring with an α-α connection, which is the C-H out-of-plane bending vibration in the pyrrole [52]. In the instance of SiO_2_-rGO@PPy nanocomposite, the peaks of SiO_2_, rGO, and PPy are plainly discernible. Consequently, it can be demonstrated that PPy grows directly on SiO_2_-rGO nanoparticles.

To examine the proportion of each component, thermogravimetric analysis (TGA) was used to determine the relative content of PPy in the SiO_2_-rGO@PPy nanocomposites (Figure 1c). The SiO_2_ curve displays one stage of weight loss (7.5 wt.%) at around 200 °C because of water extraction. Due to the residual Si-OH groups on their surfaces, SiO_2_ possesses hydrophilic properties. Then, no weight loss happens, and 92.5% of the original weight remains. Upon heating up to 1000 °C under an air atmosphere, pure PPy completely burns at a temperature of around 800 °C, leaving behind 4.2 wt.% of residue. In the case of the nanocomposite curve, the curve reveals two weightless regions when the samples were heated from 30 to 1000 °C at a rate of 10 °C/min. The first low-temperature weight loss occurred between 30 °C and ~200 °C due to the evaporation of water bound or trapped in the synthesized nanocomposite [42], while the second weight loss occurred between 200 °C and 680 °C, which can be attributed to PPy oxidation. The PPy content in the SiO_2_-rGO@PPy was calculated at 33.9% by weight. Simultaneously, the percentages of SiO_2_ and rGO were 17.4 and 48.6 wt.%, respectively.

As depicted in Figure 2, the morphological and microstructural investigation of synthesized materials was accomplished by obtaining SEM and TEM images. The microstructural features of synthesized polypyrrole (PPy), as shown in Figure 2a, indicated irregular granular particles, which is consistent with prior research [48,53]. Figure 2b demonstrates a gauzy, highly wrinkly stack of ultrathin graphene oxide nanosheets decorated with SiO_2_ nanoparticles on the surface of the rGO sheets. The TEM images of the SiO_2_-rGO nanocomposites (Figure 2c) illustrate that the silica nanoparticles are evenly distributed in the wrinkly reduced graphene oxide sheet. In situ oxidative polymerization of the pyrrole monomer led to the formation and coating of PPy layers on the surface of SiO_2_-rGO. It is crucial to highlight that the PPy layers were equally dispersed on the SiO_2_-rGO surfaces, as evidenced by the SEM (Figure 2d,e) and TEM images (Figure 2f). The modification of the morphology of SiO_2_-rGO with pyrrole suggests that the nanocomposites were successfully synthesized. These findings suggest that rGO sheets can serve as a support and spacer for the decoration of SiO_2_ nanoparticles and PPy polymers. In the nanocomposite, SiO_2_-rGO serves as the template for the synthesis of SiO_2_-rGO@PPy. As the pyrrole monomer was introduced to the SiO_2_-rGO suspension, it was absorbed on the surface via π-π interactions, van der Waals force, and hydrogen bonding [54,55,56]. Figure 2d displayed a wrinkled structure with a globular cluster of the polypyrrole matrix, rGO sheet, and SiO_2_ nanoparticles. In addition, the high-magnification SEM images (Figure 2e) revealed the layer thickness for the rGO sheet and the SiO_2_ entrapment within the matrix. Noteably, as depicted in the magnified TEM image in Figure 2f, the surfaces of the SiO_2_-rGO sheet are covered with a large number of PPy nanoparticles. For the intended nanocomposite SiO_2_-rGO@PPy, one observes a wrapping morphology in which exfoliated SiO_2_-rGO sheets act as templates for PPy. This occurrence is compatible with the previously described comparable phenomena [22,39]. Additionally, elemental mapping analysis was used to investigate the required spatial distribution of various components in the SiO_2_-rGO@PPy nanocomposites (Figure 3). Si (silicon) and O (oxygen) elements, which correspond to silica nanoparticles, are primarily present in flake-sized grains; C (carbon) and N (nitrogen) elements, which belong to the PPy polymer, are uniformly distributed throughout the samples, indicating the creation of the PPy layer. These results validate the homogeneous mixing of rGO and SiO_2_ and the uniform distribution of PPy on the surface of the SiO_2_-rGO nanocomposite particles.

### 3.2. Electrochemical Properties

Lithium-ion batteries: 2016 coin-type half-cell batteries were assembled in an argon-filled glove box with SiO_2_-rGO@PPy anodes and pure Li metal as the counter electrodes. The electrodes were prepared by doctor-blading the well-mixed slurries onto copper foil as the active material. Initially, the electrochemical performance of the SiO_2_-rGO@PPy nanocomposite was investigated in a half-cell configuration using cyclic voltammetry (CV) measurements. Figure 4a represents the first three cycles in a voltage range of 0.01–3 V (vs. Li^+^/Li) at a scan rate of 0.1 mV s^−1^. On account of the degradation of the electrolyte and the formation of the solid electrolyte interphase (SEI) film, the broad reduction peaks detected below 1.2 V in the first cycle diminished in the subsequent cycles, which correlated to the initial capacity loss [57,58,59]. A further anodic peak at around 1.6 V corresponded to the alloying process of Li*_x_*Si*_y_* [1,60]. In subsequent cycles, the CV curves were nearly overlapping, signifying excellent stability and reversibility.

Figure 4b depicts the charge and discharge profiles of the SiO_2_-rGO@PPy nanocomposite electrode for the initial three cycles at a current density of 100 mA g^−1^. Initial discharge/charge cycles reveal that the charge capacity (699 mAh g^−1^) was less than the discharge capacity (1299 mAh g^−1^), resulting in an initial Coulombic efficiency of approximately 54%. This Coulombic efficiency illustrates the reversibility of the SiO_2_-rGO@PPy nanocomposite electrode and relates to the creation of a solid electrolyte interphase (SEI) between the electrolyte and electrode, in which the potential gradient plateaus around 1.5 V [26,61]. The SEI film is a layer of coverage in the passivation layer on the surface of the electrode materials. It is the reaction result of the electrode material and electrolyte at the solid–liquid interface during the first charge–discharge cycle. The enormous initial irreversible reaction in SiO_2_-based anodes is a well-known feature that leads to increased SEI layer formation. In addition, the low initial Coulombic efficiency of SiO_2_-rGO@PPy nanocomposites is owed to the lithium loss produced by the reduction of silica. It has been proposed that SiO_2_ reacts with Li^+^ as follows. The amorphous nano-SiO_2_ in the SiO_2_-rGO@PPy nanocomposites was reduced to Si and Li_2_O or Li_4_O_4_, which consumes a substantial quantity of lithium and considerably contributes to the irreversible capacity of the initial discharge [1,62]. This highly reversible reaction adds to the greater reversibility of the nanocomposite. As lithium-ion intercalation and deintercalation occur during cycling, carbon also contributes to the cyclability and stability of nanocomposites [63]. The mechanism of SiO_2_-rGO@PPy’s electrochemical reaction can be stated as follows [64]:SiO_2_ + 4Li^+^ + 4e^−^ → 2Li_2_O + Si(1)
2SiO_2_ + 4Li^+^ + 4e^−^ → Li_4_SiO_4_ + Si(2)
Si + *x*Li^+^ + *x*e^−^ → Li*_x_*Si(3)
C + *y*Li^+^ + *y*e^−^ → Li*_y_*C(4)

After the first cycle, a discharge capacity of 725.3 mAh g^−1^ was reached with a Coulombic efficiency of 90.4%. The Coulombic efficiency of the SiO_2_-rGO@PPy nanocomposite electrode was also improved, and the plateau in the curve went away. The reason may be that the charging and discharging cycles of the electrolyte gradually became stable as the protective layer lowered the electrolyte’s contact with the electrode, permitting the production of a stable SEI film [7,22,59]. The slope below 0.5 V is attributable to irreversible electrochemical reactions between lithium ions with SiO_2_ and a series of silicate salts (Equations (1) and (2)), which resulted in an increase in irreversible capacity and, consequently, a decrease in Coulombic efficiency. Lithium ions subsequently initiated a reaction with the reduced amorphous silicon, as shown in Equation (3), resulting in reversible capacity for the following cycles. Reversible capacity can likewise be provided by porous carbon (Equation (4)) [22,65]. For potentials greater than 1.5 V, all charge voltage profiles appeared to be extremely steep. On the surface of the oxide electrode, a solvent decomposition reaction may potentially contribute to the formation of the SEI layer.

Due to the fact that high-rate capability is advantageous for the design of high-power-type LIB anode materials, the great rate performance of the electrode is also a crucial factor when evaluating various practical LIB applications. As demonstrated in Figure 4c, the SiO_2_-rGO@PPy nanocomposite electrode possessed superior rate capability in comparison to SiO_2_-rGO and bare PPy electrodes. Each step included ten discharge cycles at current densities ranging from 0.1 to 2.0 A g^−1^. The discharge capacity of the SiO_2_-rGO@PPy nanocomposite was determined to be 599.0, 489.6, 425.8, 342.8, 309.7, and 223.4 mAh g^−1^ at 0.1, 0.2, 0.4, 0.8, 1, and 2 A g^−1^, respectively. In addition, when the current was reduced to 0.1 A g^−1^, the high capacity of 583.7 mAh g^−1^ was promptly restored, indicating that the SiO_2_-rGO@PPy nanocomposite electrode has a high degree of reversibility. In contrast, the SiO_2_-rGO nanocomposite and bare PPy electrodes demonstrated low capacities and rate performances. The SiO_2_-rGO nanocomposite electrode, for instance, had a discharge capacity of around 135.1 mAh g^−1^ at a high current density of 2 A g^−1^, and only recovered to approximately 252.3 mAh g^−1^ at 0.1 A g^−1^. The bare PPy electrode reached a negligible capacity of around 10.6 mAh g^−1^ at 2 A g^−1^ and only recovered to approximately 66.3 mAh g^−1^ when the current density was reset to 0.1 A g^−1^, which was significantly lower than that of the SiO_2_-rGO@PPy nanocomposite electrode. The amazingly high rate of capacity is attributable to the protective effect of the PPy coating on the SiO_2_-rGO nanocomposite, and the volume expansion was effectively reduced [66]. In addition, the surface PPy coating increased the surface electrical conductivity of the SiO_2_-rGO by providing an electron percolation path from the current collector to the entire surface area of each individual SiO_2_-rGO nanoparticle [7,61].

As an additional critical parameter to evaluate for electrode materials, the cycle stability of the SiO_2_-rGO@PPy nanocomposite, the SiO_2_-rGO nanocomposite, and bare PPy was investigated. Figure 4d demonstrates that the cycling performance of the SiO_2_-rGO@PPy nanocomposites is substantially better than that of bare PPy nanoparticles and SiO_2_-rGO nanocomposites. After 250 cycles at a current density of 100 mA g^−1^, the SiO_2_-rGO@PPy indicated remarkable cycling stability with a capacity of 534 mAh g^−1^, whereas the SiO_2_-rGO nanocomposite revealed a decreasing capacity of 201 mAh g^−1^. The SiO_2_-rGO@PPy nanocomposite electrode retained 83% of its capacity from the first cycle to 250 cycles, but the SiO_2_-rGO electrode only retained 46%. The outstanding cycling stability of the SiO_2_-rGO@PPy nanocomposite electrode suggests that the addition of PPy facilitates greater exploitation and improves cycling performance. This effect can be attributed to PPy establishing a conductive network and maintaining the entire structure by preventing the pulverization of SiO_2_ [7,8,67]. PPy could play a more significant role in the anode materials of the lithium-ion battery, according to some speculation.

Table 1 compares the electrochemical performances of the SiO_2_-rGO@PPy nanocomposite electrode to the electrochemical performances of other reported electrodes for lithium-ion batteries, indicating that the synergistic effect of PPy and SiO_2_-rGO nanocomposites provides an appropriate performance in SiO_2_-rGO@PPy nanocomposites. SiO_2_-rGO@PPy nanocomposites have a maximum discharge capacity of 534 mAh g^−1^, which is approximately 79% of their calculated theoretical capacity (662 mAh g^−1^). It demonstrates that PPy coated on SiO_2_-rGO nanocomposite can enhance the active surface and facilitate the electrochemical reaction of active materials and Li^+^ ions, thereby increasing the Li storage capacity [34].

To thoroughly explain the superior performance of the SiO_2_-rGO@PPy nanocomposite electrodes, we performed an electrochemical impedance spectroscopy (EIS) examination of the SiO_2_-rGO@PPy electrode before and after 250 cycles of discharge (Figure 5). Figure 5a depicts a simulated equivalent circuit from EIS on both materials [70]. Figure 5b illustrates the plot of real impedance (Z’) versus the reciprocal root square of the lower angular frequencies (ω), and Table 2 provides the related simulation results. Before and after cycle testing, it is evident from Figure 5a that the curve was composed of a semicircle in the high-frequency region and a line with a slope of approximately 45° in the low-frequency region. The intercept at the real axis, which corresponds to the ohmic resistance (*R*_s_), represents the electronic conductivity of the separator, electrolytes, and anodes. The radius of the semicircle describes the charge transfer resistance (*R*_ct_), whereas the sloping line in the low-frequency region represents the Warburg impedances (*σ_W_*), which are related to the diffusion of Li^+^ ions in active materials [71,72]. In general, before the cycle test, as can be seen from Table 2, the internal resistance (*R*_s_) of SiO_2_-rGO was slightly decreased by coating PPy. SiO_2_-rGO@PPy exhibited a smaller semicircle than SiO_2_-rGO, indicating that the solid-state interface layer and charge transfer resistance (*R*_ct_, 193 Ω) were lower than SiO_2_-rGO (*R*_ct_, 399 Ω). The enhancement in SiO_2_-rGO@PPy’s electrical conductivity is likely due to the resistance of the solid-state interface layer’s PPy additive. It can be determined that the addition of PPy results in the formation of a thinner interface layer and a reduction in the transfer resistance of the SiO_2_-rGO@PPy nanocomposites, which greatly increases the Li-ion transfer rate in the SiO_2_-rGO structure.

To achieve precise values of the impedances and diffusion coefficient, the oblique linear Warburg part was fitted, and the results are shown in Figure 5b (plot of Z′ vs. ω ^−0.5^). The Warburg impedance at low frequencies determines the Li-ion diffusion coefficient. Table 2 displays the Warburg coefficient *σ_W_* for the SiO_2_-rGO@PPy and SiO_2_-rGO electrodes (the slope of the Z′ against ω^−0.5^ plot). Consequently, the lithium-ion diffusion coefficient can be calculated using Equation (5) [73,74]: (5)$$D_{{Li}^{+}}=\frac{R^{2}T^{2}}{{2A}^{2}n^{4}F^{4}C^{2}\sigma_{W^{2}}}$$
where $D_{{Li}^{+}}$ is the diffusion coefficient of lithium ions, *R* is the gas constant (8.314 J K^−1^ mol^−1^), *T* is the room temperature in our experiment (298 K), *A* is the surface area of the electrode, *n* is the number of electrons per molecule participating in the electron transfer reaction, *F* is the Faraday constant (96,500 C mol^−1^), *C* is the concentration of lithium ions, and *σ_W_* is the Warburg coefficient. The Li^+^ ion diffusion coefficients for the SiO_2_-rGO@PPy and SiO_2_-rGO electrodes are provided in Table 2 based on the previously mentioned equation. $D_{{Li}^{+}}$ of the SiO_2_-rGO@PPy electrode was significantly more than that of SiO_2_-rGO electrode. It reveals that the SiO_2_-rGO@PPy nanocomposite structure is conductive to electrolyte diffusion and enhances Li^+^ migration during the lithiation/delithiation process. In addition, it is evident that the lithium-ion diffusion coefficient decreases after the cycle test because, during the lithium-ion insertion and extraction processes, some Li^+^ ions may become trapped in the irreversible sites of SiO_2_-rGO@PPy, blocking some Li^+^ diffusion channels. This indicates a strong relationship between the lithium-ion diffusion coefficient and the degree of lithium insertion into SiO_2_-rGO@PPy.

A detailed kinetic study is conducted for a lithium storage device in order to evaluate the possible reasons for the improved rate capability and cycling stability of the SiO_2_-rGO@PPy electrode. By evaluating CV profiles at different scan rates, the lithium storage kinetics of SiO_2_-rGO@PPy electrodes were studied. It is possible to examine the high-rate performance of SiO_2_-rGO@PPy electrodes by evaluating the capacitance contribution. In the meantime, the influence of SiO_2_-rGO complexation in PPy was further investigated. To further comprehend the electrochemical kinetics of the SiO_2_-rGO@PPy electrode, cyclic voltammetry investigations were performed. The CV curves at various scan rates ranging from 0.1 to 2 mV s^−1^ contained comparable shapes, and the peaks became progressively larger as the scan rate increases (Figure 6a). The total stored charge could be contributed to in several different ways, and can be described by examining the CV data at various scan rates in accordance with equations [75,76]:*i* = *av^b^*(6)
log(*i*) = *b*log(*v*) + log(*a*)(7)
where *i* denotes the magnitude of the current, *v* denotes the scan rate, and *a* and *b* are variables. The capacity contributed by the capacitive effect could be estimated by drawing log(*i*)-log(*v*) curves based on Equation (6) and fitting the line to find its slope (the slope is the value of *b*). Previous studies have reported that when *b* approaches 0.5, a process is governed by complete diffusion, and when *b* tends to 1, the process is capacitive [75,77,78]. Consequently, by determining the value of *b*, the main contribution of the battery capacity could well be quantified. The *b* value was 0.91 at peak A (charge) and 0.99 at peak B (discharge). Figure 6b presents the linear fit. This value revealed that the capacitive and diffusion-controlled processes exert nearly equal control over the peak current. The overall capacitive contribution at a particular scan rate can be quantified by splitting the specific contribution from the capacitive and diffusion-controlled processes at a defined voltage according to the following equations [79,80]:*i* = *k*_1_*v* + *k*_2_*v*^1/2^(8)
*i/v*^1/2^ = *k*_1_*v*^1/2^ + *k*_2_(9)
in which *v* is the scan rate at certain potentials, and *k*_1_ and *k*_2_ are constants for a given potential that could be determined by linearly fitting *i*/*v*^1/2^ versus *v*^1/2^ under the defined potentials. Capacitive and diffusion contributions can be found by determining *k*_1_ as the slope and *k*_2_ as the intercept; this provides capacitive and diffusion contributions. Comparing the shaded area (*k*_1_*v*) with the experimental currents (solid line) in Figure 6c reveals that capacitive processes contributed approximately 96.7% of the total current of the SiO_2_-rGO@PPy electrode at 2 mV s^−1^. Consequently, contribution ratios between the two different ways were also determined at various scan rates. The percentage of capacitive contribution greatly increases with scan rate, as shown in Figure 6d. According to the current separation approach, capacitive contributions significantly rose from 87% to 90%, 94%, 95%, 96%, and 97%, respectively, of the overall capacitive contributions when the scan rate increased through 0.1, 0.2, 0.5, 1.0, 1.5, and 2.0 mV s^−1^. These findings revealed that the capacitive-controlled process accounted for a significant amount of the whole electrochemical process of the SiO_2_-rGO@PPy electrode. For this reason, the large capacitive contribution to the rate performance of the SiO_2_-rGO@PPy electrode showed that the material has efficient redox reactions and activity that is not affected by the rate of electrochemical reactions. Moreover, this can also imply that the conductive PPy networks in the SiO_2_-rGO@PPy electrode are of the desired quality and maintain their stable structure well during charge–discharge processes [81,82].

The CV curve of the SiO_2_-rGO@PPy electrode at 2.0 mV s^−1^ (Figure 6c) was utilized to indicate a deviation of *i* vs. $1/\sqrt{t}$, as depicted in Figure 7. This deviation suggests that the electrode exhibits either strictly linear or approximately linear behavior. The sloped (*k*) can be used to accurately calculate the Li^+^ diffusion coefficient ($D_{{Li}^{+}}$) using the Cottrell equation, as shown in Equation (10):(10)$$i=\frac{nFAC_{0}\sqrt{D_{Li}}}{\sqrt{\pi t}}$$
where *i* is the current in A, *n* is the number of electrons in the reduction or oxidation reaction of the analyte, *F* is the Faraday constant, 96,485 C/mol, *A* is the area of the planar electrode in cm^2^, *C*_0_ is the initial concentration of the analyte being reduced or oxidized (mol/dm^3^), $D_{Li}$ is the diffusion coefficient for the analyte in cm^2^/s, and *t* is the time in seconds (s).

The fundamental equation governing the Cottrell equation, when implemented, characterizes the current degradation of a planar electrode. Experimental characterization involves the simplification of the Cottrell equation as follows:(11)$$i=k\frac{1}{\sqrt{t}}$$
where
(12)$$k=\frac{nFAC_{0}\sqrt{D_{Li}}}{\sqrt{\pi }}$$

By applying this simplification, redox activities related to secondary processes, such as ligand attachment, dissociation, and conformational changes, can be distinguished through the observation of linearity deviations in the plot of *i* vs. $1/\sqrt{t}$.

The process of determining the slope (*k*) through the application of a straight line fitting to the data points during a certain time is depicted in Figure 7. Equations (11) and (12) were utilized in the complete parameter study to compute the $D_{{Li}^{+}}$ value of the assembled cell from the SiO_2_-rGO@PPy electrode. At a scan step of 2.0 mV s^−1^, the $D_{{Li}^{+}}$ of the SiO_2_-rGO@PPy electrode was calculated to be 1.51 × 10^−17^ cm^2^ s^−1^. The discrepancy between the results calculated from the Cottrell equation (1.5073 × 10^−17^ cm^2^ s^−1^) and the value obtained from the EIS discussion section (7.04 × 10^−16^ cm^2^ s^−1^), using the relationship between the real impedance (Z′) and reciprocal square root of frequencies (ω^−0.5^) for the SiO_2_-rGO@PPy electrode after cycling, is immediately evident. The two derived values are exceptionally similar. Furthermore, this discovery elucidates and verifies the precise $D_{{Li}^{+}}$ value and associated Li^+^ activities of the prepared SiO_2_-rGO@PPy electrode, which align with the battery performances that were previously deliberated.

## 4. Conclusions

The synthesis of SiO_2_-rGO@PPy nanocomposites was successfully achieved, demonstrating their novel characteristics as high-performance anode materials. In conclusion, new nanostructured rice husk-derived nano SiO_2_-rGO@PPy ternary nanocomposites were successfully synthesized through in situ chemical polymerization of the pyrrole monomer and demonstrated to be a promising anode material for lithium-ion batteries. After being coated with conductive PPy, the binary SiO_2_-rGO nanocomposites exhibited a high capacity of over 524 mAh g^−1^, long-term cycling with over 250 cycles at 100 mA g^−1^, and a good rate performance. The results demonstrate that rice husk-derived nano-SiO_2_ were successfully immobilized in rGO via PPy wrapping. In addition, the unique properties of the conductive polypyrrole coatings are responsible for the improved electrochemical performance. First, polypyrrole has excellent conductivity and increases the nanocomposites’ conductivity during the charging–discharging process. Second, the polypyrrole surface acts as a cushion to restrain the expansion of silica nanoparticles. Our proposed methodology exhibits potential for the fabrication of nanocomposites, including surface structures composed of conducting polymers. This topic is anticipated to generate significant attention among a wide range of individuals and is expected to have far-reaching implications across various disciplines. The specialty of sustainable nanocomposite development is an exciting prospect of investigation, with the potential to address significant challenges associated with lithium-ion batteries, including their manufacturing feasibility and industrial scalability. This is due to their sustainability and reduced environmental effect.

## Data Availability

The data presented in this study are available upon request from the corresponding author.

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
