# Peer review of "Insight into the Role of Conductive Polypyrrole Coated on Rice Husk-Derived Nanosilica-Reduced Graphene Oxide as the Anodes: Electrochemical Improvement in Sustainable Lithium-Ion Batteries"

_polymers, 2023, doi:10.3390/polym15244638_

Round 1

Reviewer 1 Report

Comments and Suggestions for Authors

This work is devoted to the graphene oxide nanocomposite with reduced silica content (SiO2-rGO@PPy), coated with polypyrrole derived from rice husk (SiO2-rGO@PPy) as anode material was developed using a simple composite technology followed by in-situ polymerization process.

1.Literature review should be supplemented with an evaluation of methods based on electrochemical approach for electrode preparation [Shchegolkov, A.V., Lipkin, M.S. & Shchegolkov, A.V. Preparation of WO3 Films on Titanium and Graphite Foil for Fuel Cell and Supercapacitor Applications by Electrochemical (Cathodic) Deposition Method. Russ J Gen Chem 92, 1161-1167 (2022). https://doi.org/10.1134/S1070363222060317]

2.Figures 1,4 and 6 are of poor quality. Their resolution should be improved.

3. the conclusions and abstract should be improved in terms of the scientific novelty of the work.

4. It is necessary to add a comparative table with the data of other authors.

5. Organize the list of references in accordance with the requirements of the journal.

Comments on the Quality of English Language

Minor editing of English language required

Author Response

ANSWERS TO EDITORS AND REVIEWERS COMMENTS

Manuscript ID: polymers-2732785

TITLE:

Insight into the Role of Conductive Polypyrrole Coated on Rice Husk-Derived Nanosilica-Reduced Graphene Oxide as the Anodes: Electrochemical Improvement in Sustainable Lithium-ion Batteries

AUTHORS:

Natthakan Ratsameetammajak, Thanapat Autthawong, Kittiched Khunpakdee, Mitsutaka Haruta, Torranin Chairuangsri, and Thapanee Sarakonsri

1. Summary

Thank you for the opportunity to submit a revised draft of my manuscript to the Special Issue “Advanced in Polymer/Graphene Composites and Nanocomposites” of the open access journal Polymers, titled “Insight into the Role of Conductive Polypyrrole Coated on Rice Husk-Derived Nanosilica-Reduced Graphene Oxide as the Anodes: Electrochemical Improvement in Sustainable Lithium-ion Batteries” (manuscript ID: polymers-2732785). On behalf of the aforementioned co-authors, I would like to express our gratitude to both the editors and reviewers for devoting their significant time to reviewing our manuscript. Their comments were incredibly beneficial. We have been able to incorporate changes to reflect most of the suggestions provided by the reviewers. We have highlighted the changes within the manuscript.

Here is a point-by-point response to the reviewers’ comments and concerns.

We look forward to hearing from you in due time regarding our submission and to responding to any further questions and comments you may have.

Yours respectfully

Dr. Thapanee Sarakonsri

2. Questions for General Evaluation

Reviewer’s Evaluation

Response and Revisions

Does the introduction provide sufficient background and include all relevant references?

Can be improved

Your assistance with the manuscript that you suggested is greatly appreciated. Already, we have enhanced the introduction by incorporating every relevant reference and offering appropriate context.

Are all the cited references relevant to the research?

Must be improved

We enormously value the support you provided regarding the manuscript you recommended. All cited references that are appropriate to the research have been improved thus far.

Is the research design appropriate?

Must be improved

We greatly appreciate the assistance that you rendered with respect to the manuscript that you suggested. Thus far, the research design that is suitable for the investigation has been enhanced.

Are the methods adequately described?

Can be improved

We wish to express our sincere gratitude for the assistance you provided in regards to the manuscript recommendation. The methodologies that have been described adequately at this point have been improved.

Are the results clearly presented?

Must be improved

We would like to extend our deepest thanks for the support that you offered regarding the manuscript recommendation. The outcomes reported have improved.

Are the conclusions supported by the results?

Can be improved

On behalf of the manuscript recommendation team, we wish to express our profoundest gratitude for your assistance. The results provided greater support for the conclusions reached.

3. Point-by-point response to Comments and Suggestions for Authors

Comments 1: Literature review should be supplemented with an evaluation of methods based on electrochemical approach for electrode preparation [Shchegolkov, A.V., Lipkin, M.S. & Shchegolkov, A.V. Preparation of WO3 Films on Titanium and Graphite Foil for Fuel Cell and Supercapacitor Applications by Electrochemical (Cathodic) Deposition Method. Russ J Gen Chem 92, 1161-1167 (2022). https://doi.org/10.1134/S1070363222060317]

Response 1: We greatly appreciate your assistance with the suggested manuscript. We have conducted a comprehensive examination and given due consideration to the references provided by the reviewer. Despite the significance of the information provided, we had trouble appropriately integrating it within the context of our study.

Comments 2: Figures 1,4 and 6 are of poor quality. Their resolution should be improved.

Response 2: You have brought forth a highly important observation. In order to enhance the resolution, replacements have been done for Figures 1, 4, and 6.

Comments 3: the conclusions and abstract should be improved in terms of the scientific novelty of the work.

Response 3: Your observation is greatly appreciated. Your suggestion has been integrated into the entire revised manuscript in accordance with our agreement with this remark. The abstract and conclusions pertaining to the scientific novelty of the work were presented in this section.

Abstract: “Polypyrrole (PPy) is a type of conducting polymer that has garnered attention as a potential electrode material for sustainable energy storage devices. This is mostly attributed to its mechanical flexibility, ease of processing, and ecologically friendly nature. Here, polypyrrole-coated rice husk-derived nanosilica-reduced graphene oxide nanocomposite (SiO2-rGO@PPy) as an anode material has been developed by a simple composite technique followed by an in-situ polymerization process. The architecture of reduced graphene oxide offers a larger electrode/electrolyte interface to promote charge-transfer reactions and provides sufficient space to buffer a large volume expansion of SiO2, maintaining the mechanical integrity of the overall electrode during the lithiation/delithiation process. Moreover, the conducting polymer coating not only improves the capacity of SiO2, but also suppresses the volume expansion and rapid capacity fading caused by serious pulverization. The present anode material shows a remarkable specific reversible capacity of 523 mAh g-1 at 100 mA g-1 current density and exhibits exceptional discharge rate capability. The cycling stability at a current density of 100 mA g-1 shows 81.6% capacity retention and high coulombic efficiency after 250 charge-discharge cycles. The study also pointed out that this method might be able to be used on a large scale in the lithium-ion battery industry, which could have a big effect on its long-term viability. Creating sustainable nanocomposites is an exciting area of research that could help solve some of the biggest problems with lithium-ion batteries, like how easy they are to make and how big they can be used in industry. This is because they are sustainable and have less of an impact on the environment.”

Conclusions: “The synthesis of SiO2-rGO@PPy nanocomposites has been successfully achieved, demonstrating their novel characteristics as high-performance anode materials. In conclusion, new nanostructured rice husk-derived nano SiO2-rGO@PPy ternary nanocomposites have been successfully synthesized through in-situ chemical polymerization of pyrrole monomer and demonstrated to be a promising anode material for lithium-ion batteries. After being coated with conductive PPy, the binary SiO2-rGO nanocomposites exhibited a high capacity of over 524 mAh g-1, long-term cycling with over 250 cycles at 100 mA g-1, and good rate performance. The results demonstrate that rice husk-derived nano-SiO2 has been successfully immobilized in rGO via PPy wrapping. In addition, the unique properties of the conductive polypyrrole coatings are responsible for the improved electrochemical performance. First, polypyrrole has excellent conductivity and increases the nanocomposites’ conductivity during the charging-discharging process. Second, the polypyrrole surface acts as a cushion to restrain the expansion of silica nanoparticles. Our proposed methodology exhibits potential for the fabrication of nanocomposites, including surface structures composed of conducting polymers. This topic is anticipated to generate significant attention among a wide range of individuals and is expected to have far-reaching implications across various disciplines. The specialty of sustainable nanocomposite development is an exciting prospect of investigation with the potential to address significant challenges associated with lithium-ion batteries, including their manufacturing feasibility and industrial scalability. This is due to their sustainability and reduced environmental effects.”

The detailed explanations can be found in the highlighted part of the revised manuscript, specifically on pages 1 and 12.

Comments 4: It is necessary to add a comparative table with the data of other authors.

Response 4: In accordance with the reviewer’s suggestion, include a comparative table with data from several authors. Consequently, this comparative table is presented in the manuscript as Table 1 (on page 8).

Comments 5: Organize the list of references in accordance with the requirements of the journal.

Response 5: Your attention to this is greatly appreciated. The list of references has been modified to meet the requirements of the journal in the revised manuscript accordingly.

4. Response to Comments on the Quality of English Language

Point 1: Minor editing of English language required

Response 1: I appreciate your attention to this matter. The majority of the manuscripts have demonstrated that the English language is of higher quality.

Reviewer 2 Report

Comments and Suggestions for Authors

This article reports on preparation and characterization of polypyrrole coated rice husk derived nanosilica-reduced graphene oxide nanocoposite (SiO2-rGO@PPy) as anode material for lithium ion batteries. The work is interesting because nanocomposite materials are extensively researched for application in energy storage and conversion devices. The manuscript is a good written scientific document and is publishable in Polymers after major revision in the light of following comments:

1. The introduction needs to be improved by highlighting the role of conducting polymers generally and polypyrrole specifically for application in energy storage devices such as batteries and ultracapacitors. 

2. No information is presented on the extraction of SiO2 from rice husk. It is not clear whether they have extracted SiO2 from rice husk or readily purchased.

3. Describe method of preparation of reduced graphene oxide in the revised manuscript.

4. Revise scheme 1 in the light of comments 2 and 3.

5. Describe the method used for loading to active materials on working electrode.

6. It will be more appropriate to determine Diffusion coefficient from cyclic voltammetry data using Cottrell’s equations and compare the findings with the value determined from EIS measurements.

7. Incorporate error bars in figure 6b.

8. They should also calculate specific capacitances from charge discharge curves and compare with those found from cyclic voltametric data.

9. The stability of the anode materials should be checked at least for 1000 cycles. The have recorded only 250 cycles.

10. Correct several tyops and deleted repeating sentenses.

Comments on the Quality of English Language

Overall the English language is satisfactory. Minor editing will be required.

Author Response

ANSWERS TO EDITORS AND REVIEWERS COMMENTS

Manuscript ID: polymers-2732785

TITLE:

Insight into the Role of Conductive Polypyrrole Coated on Rice Husk-Derived Nanosilica-Reduced Graphene Oxide as the Anodes: Electrochemical Improvement in Sustainable Lithium-ion Batteries

AUTHORS:

Natthakan Ratsameetammajak, Thanapat Autthawong, Kittiched Khunpakdee, Mitsutaka Haruta, Torranin Chairuangsri, and Thapanee Sarakonsri

1. Summary

Thank you for the opportunity to submit a revised draft of my manuscript to the Special Issue “Advanced in Polymer/Graphene Composites and Nanocomposites” of the open access journal Polymers, titled “Insight into the Role of Conductive Polypyrrole Coated on Rice Husk-Derived Nanosilica-Reduced Graphene Oxide as the Anodes: Electrochemical Improvement in Sustainable Lithium-ion Batteries” (manuscript ID: polymers-2732785). On behalf of the aforementioned co-authors, I would like to express our gratitude to both the editors and reviewers for devoting their significant time to reviewing our manuscript. Their comments were incredibly beneficial. We have been able to incorporate changes to reflect most of the suggestions provided by the reviewers. We have highlighted the changes within the manuscript.

Here is a point-by-point response to the reviewers’ comments and concerns.

We look forward to hearing from you in due time regarding our submission and to responding to any further questions and comments you may have.

Yours respectfully

Dr. Thapanee Sarakonsri

2. Questions for General Evaluation

Reviewer’s Evaluation

Response and Revisions

Does the introduction provide sufficient background and include all relevant references?

Must be improved

Your assistance with the manuscript that you suggested is greatly appreciated. Already, we have enhanced the introduction by incorporating every relevant reference and offering appropriate context.

Are all the cited references relevant to the research?

Can be improved

We enormously value the support you provided regarding the manuscript you recommended. All cited references that are appropriate to the research have been improved thus far.

Is the research design appropriate?

Can be improved

We greatly appreciate the assistance that you rendered with respect to the manuscript that you suggested. Thus far, the research design that is suitable for the investigation has been enhanced.

Are the methods adequately described?

Must be improved

We wish to express our sincere gratitude for the assistance you provided in regards to the manuscript recommendation. The methodologies that have been described adequately at this point have been improved.

Are the results clearly presented?

Can be improved

We would like to extend our deepest thanks for the support that you offered regarding the manuscript recommendation. The outcomes reported have improved.

Are the conclusions supported by the results?

Can be improved

On behalf of the manuscript recommendation team, we wish to express our profoundest gratitude for your assistance. The results provided greater support for the conclusions reached.

3. Point-by-point response to Comments and Suggestions for Authors

Comments 1: The introduction needs to be improved by highlighting the role of conducting polymers generally and polypyrrole specifically for application in energy storage devices such as batteries and ultracapacitors.

Response 1: An essential observation has been made regarding the necessity of enhancing the introduction section by emphasizing the significance of conducting polymers in general, and specifically polypyrrole, for their potential application in energy storage systems. The result was explained in this part.

“Conducting polymers (CPs) have a distinctive structural feature, wherein their polymer backbone is composed of a sequence of alternating single and double bonds. The system of delocalized π-electrons is generated by the overlapping p-orbitals, resulting in optical and electrical features that are both fascinating and valuable [1]. According to recent research, it has been shown that these materials generally demonstrate an electrical conductivity that varies within the range of 0.01 to 500 S cm-1 [2]. Additionally, there has been a growing interest in electrochemical energy storage devices that possess qualities like as strength, flexibility, low production costs, low self-discharge rates, tolerance to overdischarge, and long cycle life [3]. Polyheterocycles, including polypyrrole (PPy), polyaniline (PANI), polythiophene (PTh), poly(3,4-ethylenedioxythiophene) (PEDOT), and their derivatives [4], are considered to be highly intriguing conducting polymers (CPs) in the context of battery applications. These materials have been the subject of investigation due to their exceptional stability in atmospheric conditions and favorable electrochemical characteristics. These materials have been the subject of investigation due to their exceptional stability in atmospheric conditions and favorable electrochemical characteristics. PPy has garnered significant interest within the field of conductive polymers because to its exceptional attributes. These include its comparatively affordable cost, remarkable redox capabilities, commendable electrical conductivity, biocompatibility, and chemical stability [5–7].”

The detailed explanations can be found in the part that is highlighted in yellow of the revised manuscript. (on page 2)

Comments 2: No information is presented on the extraction of SiO2 from rice husk. It is not clear whether they have extracted SiO2 from rice husk or readily purchased.

Response 2: The recommendation provided has been incorporated into the updated article in its entirety, as it aligns with our agreement. This response presents the process of extracting SiO2 from rice husks.

“The rice husks (RHs) obtained from agricultural waste were subjected to a rinsing process using water at room temperature in order to eliminate any mechanical impurities and dust. Subsequently, the RHs were dried at a temperature of 100 °C until they reached a state of total dryness. The RHs that had been rinsed and dried were immersed in an aqueous solution of HCl overnight, with the purpose of removing any metal impurities and promoting the hydrolysis of cellulose and hemicellulose. The sample was subsequently rinsed with water until a neutral pH of 7 was achieved, followed by drying at a temperature of 100 °C. Subsequently, the RHs underwent a two-stage burning process in a muffle furnace: first at a temperature of 500°C for a duration of 2 hours, followed by a subsequent burning at a temperature of 700 °C for a duration of 2 hours.”

A more comprehensive exposition of these explanations can be found in the part that is highlighted in light blue (on Page 3) and Scheme 1 of the revised manuscripts (on Page 4).

Comments 3: Describe method of preparation of reduced graphene oxide in the revised manuscript.

Response 3: We sincerely appreciate your notifying us of this. The process employed to generate reduced graphene oxide, which was annotated in this section, is further thoroughly illustrated in Scheme 1 of the revised manuscript (on Page 4) and the area of the amended manuscript that has been marked with yellow highlighting (on Page 3).

“Graphene oxide (GO) was synthesized from graphite powder employing a modified Hummers’ method, followed by successive reduction within a tube furnace. Reduced graphene oxide (rGO) was synthesized by subjecting it to a five-hour heat treatment at a temperature of 800 °C under an inert atmosphere.”

Comments 4: Revise scheme 1 in the light of comments 2 and 3.

Response 4: We agree with your viewpoint, and we have integrated your suggestion into the revised version of the manuscript. The updated scheme 1 was also demonstrated in this context.

The aforementioned scheme is expounded upon extensively in the designated portion of the revised manuscript, which has been visually marked with the color green. (on page 4)

Comments 5: Describe the method used for loading to active materials on working electrode.

Response 5: We express our sincere gratitude for raising this matter. This response provides a description of the methodology employed for the loading of active materials onto the working electrode. The fabrication process of the anode electrode involved the combination of active material powder, super P, and sodium alginate in a weight ratio of 70:15:15. Deionized water was utilized as the solvent during this procedure. The slurry consisting of various components was applied onto a copper foil substrate using the doctor-blade technique, followed by a drying process at a temperature of 60 °C within an oven for a duration of 12 hours.

Comments 6: It will be more appropriate to determine Diffusion coefficient from cyclic voltammetry data using Cottrell’s equations and compare the findings with the value determined from EIS measurements.

Response 6: I appreciate your suggestion. Based on your idea of using Cottrell's equations to figure out the diffusion coefficient using cyclic voltammetry data and comparing it to the value found through electrochemical impedance spectroscopy (EIS) measurement, we agree with what you say. However, we regret to inform you that we are unable to provide an explanation or engage in a discussion on this topic within the scope of the revised manuscript. Given the necessity to engage with Cottrell's equation, I managed to determine just enough time to formulate a response. Furthermore, we intend to thoroughly examine and analyze the subject matter in order to apply the acquired knowledge towards the advancement of our future projects.

Comments 7: Incorporate error bars in figure 6b.

Response 7: I would like to express my gratitude for your valuable support with regard to the manuscript that you suggested. Based on the data presented in Figure 6b, it is derived from a single half-coin cell, hence preventing the inclusion of an error bar. Due to limitations related to the availability of materials, resources, equipment, and instruments, conducting the test on several cells is not feasible. In order to gather additional data for the creation of a graph with error bars, we will attempt to conduct experiments on an average of three to five half-coin cells if we continue with the current work.

Comments 8: They should also calculate specific capacitances from charge discharge curves and compare with those found from cyclic voltametric data.

Response 8: We sincerely appreciate your initiative in raising this matter. It is advisable to compare the specific capacitance determined using both the charge-discharge curve and cyclic voltametric data, as the reviewer advised.  Nevertheless, after careful consideration and careful evaluation, I have reached the conclusion that it would be insufficient to carry out the comparison. This decision is based on the fact that such a study would exceed the scope of our manuscript, which mainly focuses on the stable specific capacity (mAh/g) rather than the specific capacitance (F/g). Since the cyclic voltametric experiment was conducted using a fixed voltage scan rate, following the charge-discharge experiment that was performed under constant current conditions, the obtained specific capacitance values exhibited variability, proving them incomparable. In addition, the charge-discharge curves demonstrate a consistent capacity during a longer cycle life. However, it is important to note that the cyclic voltametric analysis was conducted solely for three cycles, wherein the initial three cycles exhibited a notably higher capacity. If we proceed with the second work, we will contemplate the possibility of conducting an investigation into this examination afterwards.

Comments 9: The stability of the anode materials should be checked at least for 1000 cycles. The have recorded only 250 cycles.

Response 9: We express our gratitude for your effort in bringing this matter to our attention. We agree with the statement mentioned. Due to limitations involving materials, resources, equipment, and instruments, we performed a smaller number of cycles, specifically 250, as opposed to a larger number. Furthermore, the stability of the specific capacity of the anode materials under examination has been observed for 250 cycles. It is anticipated that this stability will persist in subsequent cycles until the end of the cells. Nevertheless, in the event that the experiment continues utilizing the same half-coin cells, it is expected that it will be discontinued. Conversely, if the experiment is conducted employing the new half-coin cells, it is projected that they will be utilized for an extended duration exceeding three months, especially up to 1000 cycles.

Comments 10: Correct several types and deleted repeating sentenses.

Response 10: You have raised a vital aspect. Most of the sentences were accurate, and duplicate sentences have been removed.

4. Response to Comments on the Quality of English Language

Point 1: Overall the English language is satisfactory. Minor editing will be required.

Response 1: I appreciate your attention to this matter. The majority of the manuscripts have demonstrated that the English language is of higher quality.

Round 2

Reviewer 1 Report

Comments and Suggestions for Authors

Accept in present form

Author Response

ANSWERS TO EDITORS AND REVIEWERS’ COMMENTS

Manuscript ID: polymers-2732785

TITLE:

Insight into the Role of Conductive Polypyrrole Coated on Rice Husk-Derived Nanosilica-Reduced Graphene Oxide as the Anodes: Electrochemical Improvement in Sustainable Lithium-ion Batteries

AUTHORS:

Natthakan Ratsameetammajak, Thanapat Autthawong, Kittiched Khunpakdee, Mitsutaka Haruta, Torranin Chairuangsri, and Thapanee Sarakonsri

1. Summary

Thank you for the opportunity to submit a revised draft of my manuscript to the Special Issue “Advanced in Polymer/Graphene Composites and Nanocomposites” of the open access journal Polymers, titled “Insight into the Role of Conductive Polypyrrole Coated on Rice Husk-Derived Nanosilica-Reduced Graphene Oxide as the Anodes: Electrochemical Improvement in Sustainable Lithium-ion Batteries” (manuscript ID: polymers-2732785). On behalf of the aforementioned co-authors, I would like to express our gratitude to both the editors and reviewers for devoting their significant time to reviewing our manuscript. Their comments were incredibly beneficial. We have been able to incorporate changes to reflect most of the suggestions provided by the reviewers. We have highlighted the changes within the manuscript.

Here is a point-by-point response to the reviewers’ comments and concerns.

We look forward to hearing from you in due time regarding our submission and to responding to any further questions and comments you may have.

Yours respectfully

Dr. Thapanee Sarakonsri

2. Point-by-point response to Comments and Suggestions for Authors

Comments 1: Accept in present form

Response 1: We would like to thank Reviewers for taking the necessary time and effort to review the manuscript.

Reviewer 2 Report

Comments and Suggestions for Authors

 The authors have complied with most of the earlier comments. Though the quality of manuscript has been improved but incorporation of error bars in figure 6b or calculation of diffusion coefficient from cyclic voltammetry data using Cottrell’s equation will make the manuscript more attractive for readers.

Comments on the Quality of English Language

Minor editing of English language will be required.

Author Response

ANSWERS TO EDITORS AND REVIEWERS’ COMMENTS

Manuscript ID: polymers-2732785

TITLE:

Insight into the Role of Conductive Polypyrrole Coated on Rice Husk-Derived Nanosilica-Reduced Graphene Oxide as the Anodes: Electrochemical Improvement in Sustainable Lithium-ion Batteries

AUTHORS:

Natthakan Ratsameetammajak, Thanapat Autthawong, Kittiched Khunpakdee, Mitsutaka Haruta, Torranin Chairuangsri, and Thapanee Sarakonsri

1. Summary

Thank you for the opportunity to resubmit a revised draft of my manuscript to the Special Issue “Advanced in Polymer/Graphene Composites and Nanocomposites” of the open access journal Polymers, titled “Insight into the Role of Conductive Polypyrrole Coated on Rice Husk-Derived Nanosilica-Reduced Graphene Oxide as the Anodes: Electrochemical Improvement in Sustainable Lithium-ion Batteries” (manuscript ID: polymers-2732785). On behalf of the aforementioned co-authors, I would like to express our gratitude to both the editors and reviewers for devoting their significant time to reviewing our manuscript. Their comments were incredibly beneficial. We have been able to incorporate changes to reflect most of the suggestions provided by the reviewers. We have highlighted the changes within the manuscript.

Here is a point-by-point response to the reviewers’ comments and concerns.

We look forward to hearing from you in due time regarding our submission and to responding to any further questions and comments you may have.

Yours respectfully

Dr. Thapanee Sarakonsri

2. Point-by-point response to Comments and Suggestions for Authors

Point 1: The authors have complied with most of the earlier comments. Though the quality of manuscript has been improved but incorporation of error bars in figure 6b or calculation of diffusion coefficient from cyclic voltammetry data using Cottrell’s equation will make the manuscript more attractive for readers.

Response 1: I express my gratitude for your perceptive recommendation. We wholeheartedly concur with this recommendation, as the reviewer suggested. In order to enhance the clarity and appeal of the manuscript for readers, we have computed an additional diffusion coefficient from cyclic voltammetry data, as illustrated below, utilizing Cottrell's equation. As demonstrated below, this calculation has already been discussed and added to this article (highlighted in yellow on page 13).

“The CV curve of the SiO2-rGO@PPy electrode at 2.0 mV s-1 (Figure 6c) was utilized to indicate a deviation of i vs , as depicted in Figure 7. This deviation suggests that the electrode exhibits either strictly linear or approximately linear behavior. The sloped (k) can be used to accurately calculate the Li+diffusion coefficient ( ) using the Cottrell equation, as shown in Equation 10.

(10)

Where: i is the current in A, n is the number of electrons in the reduction or oxidation reaction of the analyte, F is the Faraday constant, 96485 C/mol, A is the area of the planar electrode in cm2, C0 is the initial concentration of the analyte being reduced or oxidized (mol/dm3),  is the diffusion coefficient for the analyte in cm2/s, and t is the time in seconds (s).

The fundamental equation governing the Cottrell equation, when implemented, characterizes the current degradation of a planar electrode. Experimental characterization involves the simplification of the Cottrell equation as follows:

(11)

Where:

(12)

By applying this simplification, redox activities related to secondary processes, such as ligand attachment, dissociation, and conformational changes, can be distinguished through the observation of linearity deviations in the plot of i vs .

The process of determining the slope (k) through the application of a straight line fitting to the data points during a certain time is depicted in Figure 7. Equations 11 and 12 are utilized in the complete parameter study to compute the  value of the assembled cell from the SiO2-rGO@PPy electrode. At a scan step of 2.0 mV s-1, the  of the SiO2-rGO@PPy electrode is calculated to be 1.51×10-17 cm2 s-1. The discrepancy between the results calculated from the Cottrell equation (1.5073×10-17 cm2 s-1) and the value obtained from the EIS discussion section (7.04 x 10-16 cm2 s-1) using the relationship between real impedance (Z') and reciprocal square root of frequencies (ω-0.5) for the SiO2-rGO@PPy electrode after cycling is immediately evident. The two derived values are exceptionally similar. Furthermore, this discovery elucidates and verifies the precise  value and associated Li+ activities of the prepared SiO2-rGO@PPy electrode, which align with the battery performances that were previously deliberated.”

Additionally, as per your suggestion, the author should provide information regarding the linear fitting of charge and discharge peak current (i) with scan rate (v) using the incorporation error bars in Figure 6b. Indeed, this information was gathered and computed using a single cell. As a result, the incorporation error bars could not be obtained for inclusion in this graph, per our assessment. However, by implementing this recommendation, we will be able to improve our work in subsequent investigations and will conduct more trials until we determine the optimal experimental conditions and statistical value.

Figure 7. The line relationship of i vs 1/  and its linear fitting 

Point 2: Overall, the English language is satisfactory. Minor editing will be required.

Response 2: Your acknowledgment of a crucial aspect is deeply appreciated. The grammatical and typographical errors that were identified in the majority of the manuscripts have already been rectified in order to improve the quality of the English language.
